# Gut Microbiota Dysbiosis Ameliorates in LNK-Deficient Mouse Models with Obesity-Induced Insulin Resistance Improvement

**DOI:** 10.3390/jcm12051767

**Published:** 2023-02-22

**Authors:** Jingbo Chen, Jiawen Xu, Yan Sun, Yuhuan Xue, Yang Zhao, Dongzi Yang, Shuijie Li, Xiaomiao Zhao

**Affiliations:** 1Department of Reproductive Medicine, Guangdong Provincial People’s Hospital (Guangdong Academy of Medical Sciences), Southern Medical University, Guangzhou 510080, China; 2Department of Reproductive Medicine, Dongguan Maternal & Child Healthcare Hospital, Dongguan 523000, China; 3Department of Obstetrics and Gynecology, Sun Yat-sen Memorial Hospital, Sun Yat-sen University, Guangzhou 510275, China; 4Department of Biopharmaceutical Sciences, College of Pharmacy, Harbin Medical University, Harbin 150088, China

**Keywords:** gut microbiota, LNK deficiency, obesity, insulin resistance

## Abstract

Purpose: To investigate the potential role of gut microbiota in obesity-induced insulin resistance (IR). Methods: Four-week-old male C57BL/6 wild-type mice (*n* = 6) and whole-body SH2 domain-containing adaptor protein (LNK)-deficient in C57BL/6 genetic backgrounds mice (*n* = 7) were fed with a high-fat diet (HFD, 60% calories from fat) for 16 weeks. The gut microbiota of 13 mice feces samples was analyzed by using a 16 s rRNA sequencing analysis. Results: The structure and composition of the gut microbiota community of WT mice were significantly different from those in the LNK-/- group. The abundance of the lipopolysaccharide (LPS)-producing genus *Proteobacteria* was increased in WT mice, while some short-chain fatty acid (SCFA)-producing genera in WT groups were significantly lower than in LNK-/- groups (*p <* 0.05). Conclusions: The structure and composition of the intestinal microbiota community of obese WT mice were significantly different from those in the LNK-/- group. The abnormality of the gut microbial structure and composition might interfere with glucolipid metabolism and exacerbate obesity-induced IR by increasing LPS-producing genera while reducing SCFA-producing probiotics.

## 1. Introduction

Obesity is becoming a worldwide health risk factor, and obesity-induced morbidity and complications account for huge costs for affected individuals, families, healthcare systems, and society at large. Obesity is a low-grade sustained inflammatory state that alters the whole-body metabolism that frequently leads to insulin resistance (IR) [1], which in turn plays a vital role in the pathogenesis of obesity-associated hyperlipidemia, non-alcoholic fatty liver disease, polycystic ovary syndrome, type 2 diabetes, and atherosclerotic cardiovascular disease [2]. Nutrients and substrates as well as systems involved in host–nutrient interactions, including gut microbiota, have been also identified as modulators of metabolic pathways controlling insulin action and obesity regulation [3]. However, the molecular mechanism of IR has not been exactly clarified.

Gut microbiota is the general term for the microbes that inhabit the gastrointestinal tract of the human body. Around 98–99% of the intestinal microbiomes can be classified into four groups: Bacteroidetes, Firmicutes, Proteobacteria, and Actinomycetes. The balance of intestinal microbe species is the key to keeping the intestinal immune function normal and maintaining the homeostasis of the body. Breaking the balance will lead to serious pathophysiological changes, which is called gut microbiota dysbiosis [4]. Increasing studies showed that Bacteroides are associated with high-fat and high-protein diets [5] and the imbalance of intestinal microecology might be involved in the occurrence of many diseases, such as irritable bowel syndrome, obesity, type 2 diabetes, metabolic syndrome (MetS), and cardiovascular diseases [6,7,8].

Metagenomic sequencing and 16S RNA sequencing were used to detect the changes in intestinal microbiota in patients with prediabetes, type 2 diabetes, and MetS. Two studies found that although the races and their diets were different, in type 2 diabetes patients, the proportion of Clostridium butyrate-producing *Roche fusobacterium* and *Clostridium leptum* decreased while the proportion of non-Clostridium butyrate increased [9,10]. The levels of Firmicutes and Clostridia in the gut microbiota of type 2 diabetes patients were significantly decreased as compared to normal controls, and the ratio of Bacteroidetes to Firmicutes was increased and positively correlated with blood glucose concentrations [11]. There are changes in the intestinal microbiota in people with abnormal glucose metabolism, and the changes in the intestinal microbiota also seem to be involved in the occurrence and remission of abnormal glucose metabolism. It was reported that feces from mice with abnormal glucose metabolism transplanted into healthy germ-free mice could cause abnormal glucose metabolism [12]. Furthermore, transplanting feces from lean donors into patients with MetS could increase their gut microbiota diversity and insulin sensitivity [13]. The results above suggested that the intestinal microbiota are closely related to the occurrence and development of abnormal glucose metabolism, while IR, as an important link in the occurrence and development of abnormal glucose metabolism, also seems to be related to the intestinal microbiota.

Our previous study discovered that ovarian tissues from PCOS patients with IR exhibited higher expression of the SH2 domain-containing adaptor protein (LNK) than ovaries from normal control subjects and PCOS patients without IR [14]. In addition, we found that there were more accumulated intrahepatic triglyceride, higher serum triglyceride (TG), and free fatty acid (FFA) in wild-type (WT) mice as compared to LNK-deficient (LNK-/-) mice fed with a high-fat diet (HFD). LNK deficiency improved glucose metabolism and IR in obese mice, suggesting the LNK might play a pivotal role in controlling glucolipid metabolism and obesity-induced IR by regulating IRS1/PI3K/Akt/AS160 signaling and the AKT/FOXO3 pathway [15,16]. Therefore, we chose LNK-/- mice as the IR-improved model and WT mice as the MetS/IR model. In this study, we compared intestinal microbiota of LNK-/- mice and WT mice that consumed HFD, with the aim to explore the potential influence of gut microbiomes on the glucolipid metabolic disorder and obesity-induced IR. 

## 2. Methods

### 2.1. Animals

The study protocol was approved by the Research Ethics Board of Sun Yat-sen memorial hospital of Sun Yat-sen University and Guangdong Provincial People’s Hospital. All the experimental procedures were approved by the Committee for Animal Research of Sun Yat-sen University and the National Institutes of Health (NIH) Guide for the Care and Use of Laboratory Animals.

Four-week-old male C57BL/6 wild-type mice (*n* = 6) were purchased from the animal research center of Sun Yat-sen University. Whole-body LNK-deficient in C57BL/6 genetic backgrounds mice (*n* = 7) were created via CRISPR/Cas mediated genome engineering by Cyagen Biosciences Inc. The mouse Sh2b3 gene (GenBank accession number: NM_001306127.1; Ensembl: ENSMUSG00000042594) is located on mouse chromosome 5. Exon 1 to exon 3 were selected as target sites. Cas9 mRNA and gRNA generated using an in vitro transcription were then injected into fertilized eggs for knockout mouse production. All mice were randomly divided into different groups, housed 4 to 5 per cage, with standard laboratory conditions (12 h light:12 h darkness cycle) at a controlled temperature (23 ± 2 °C) and free access to rodent feed and water. All mice (4–5 weeks old) were fed a high-fat diet (HFD, 60% calories from fat, D12492; Research Diets Inc., New Brunswick, NJ, USA) for 16 weeks.

### 2.2. Sample Collection

When mice were fed with a HFD for up to 16 weeks, fecal samples were collected and immediately kept frozen at −80 °C until processed for analysis. Total DNA was isolated from the fecal samples using the MasterPure Complete DNA&RNA Purification Kit (Epicenter) according to the manufacturer’s instructions with some modifications as described previously [17].

### 2.3. 16S rRNA Extraction and Sequencing

DNA was extracted using a DNA extraction kit for the corresponding sample. The concentration and purity were measured using the NanoDrop One (Thermo Fisher Scientific, Waltham, MA, USA). Next, 16S rRNA/18SrRNA/ITS genes of distinct regions (e.g., Bac 16S: V3-V4/V4/V4-V5; Fug 18S: V4/V5; ITS1/ITS2; Arc 16S: V4-V5 et al.) were amplified used specific primer (e.g., 16S: 338F and 806R/515F and 806R/515F and 907R; 18S: 528F and 706R/817F and 1196R; ITS5-1737F and ITS2-2043R/ITS3-F and ITS4R; Arc: Arch519F and Arch915R et al.) with a 12bp barcode. Primers were synthesized by Invitrogen (Invitrogen, Carlsbad, CA, USA). PCR reactions, containing 25 μL 2× Premix Taq (Takara Biotechnology, Dalian Co. Ltd., Dalian, China), 1 μL each primer (10 μM), and 3 μL DNA (20 ng/μL) template in a volume of 50 µL, were amplified via thermocycling: 5 min at 94 °C for initialization; 30 cycles of 30 s denaturation at 94 °C, 30 s annealing at 52 °C, and 30 s extension at 72 °C; followed by 10 min final elongation at 72 °C. The PCR instrument was BioRad S1000 (Bio-Rad Laboratory, Hercules, CA, USA). The length and concentration of the PCR product were detected via 1% agarose gel electrophoresis. Samples with the bright main strip between (e.g., 16S V4: 290–310 bp/16S V4V5: 400–450 bp et al.) could be used for further experiments. PCR products were mixed in equidensity ratios according to the GeneTools Analysis Software (Version 4.03.05.0, SynGene, Cambridge, UK). Then, the mixture of PCR products was purified with E.Z.N.A. Gel Extraction Kit (Omega, Bellevue, WA, USA). Next, sequencing libraries were generated using NEBNext^®^ Ultra™ II DNA Library Prep Kit for Illumina^®^ (New England Biolabs, Ipswich, MA, USA) following the manufacturer’s recommendations, and index codes were added. The library quality was assessed on the Qubit@ 2.0 Fluorometer (Thermo Fisher Scientific, Waltham, MA, USA). At last, the library was sequenced on an Illumina Nova6000 platform and 250 bp paired-end reads were generated.

### 2.4. Data Analysis

Fastp (version 0.14.1) was used to control the quality of the raw data by sliding the window (-W 4 -M 20). The primers were removed by using cutadapt software according to the primer information at the beginning and end of the sequence to obtain the paired-end clean reads. Paired-end clean reads were merged using usearch -fastq_mergepairs (V10) according to the relationship of the overlap between the paired-end reads; when with at least a 16 bp overlap, the read generated from the opposite end of the same DNA fragment, the maximum mismatch allowed in the overlap region was 5 bp, and the spliced sequences were called Raw Tags. Fastp (version 0.14.1) was used to control the quality of the raw data by sliding the window (-W 4 -M 20) to obtain the paired-end clean tags.

R software was used to count the union (pan) and intersection (core) of the target classification level in different samples to evaluate whether the sample size was sufficient. R software was used to analyze the common and endemic species, the composition of the community, and the richness of species.

## 3. Results

### 3.1. Diversity Difference of Intestinal Microbiota between LNK-/- and WT Mice

A total of 13 mice (7 LNK-/-mice and 6 WT mice) were included in this study. The average body weights of 0W LNK-/-mice and WT mice were 21 g ± 2.2 g and 21.1 g ± 2.1 g, respectively, with no significance (*p* > 0.05). During the process of the mice fed with HFD, we observed that LNK-/- mice had a loss of appetite compared with WT mice. The food intakes of LNK-/- mice and WT mice were 18.8 g ± 1.3 g and 19.4 g ± 1.8 g, respectively, with statistical significance (*p* < 0.05). After 16 weeks, there was a significant difference in body weight between LNK-/- mice (47.5 g ± 4.6 g) and WT mice (52.6 g ± 3.3 g) (*p* < 0.05). All thirteen feces samples from seven LNK-/-mice and six WT mice were analyzed. A majority of intestinal microbe species of LNK-/-mice and WT mice were similar, however, the diversity of gut microbiomes in the WT mice group was less than that of the LNK-/- mice group (Figure 1A). The α diversity of the gut microbiota calculated using the Shannon index showed that the LNK-/- group species diversity was higher than that of the WT group at the phylum level (*p* < 0.05, t test) (Figure 1B,C).

### 3.2. Composition and Abundance Difference of Intestinal Microbiota between LNK-/- and WT Mice

To compare the composition difference in the intestinal microbiota between LNK-/- and WT mice, we next performed a Bray–Curtis-based principal coordinates analysis (PCoA) (Figure 2A). It was shown that the degree of similarity between the two groups of microbial communities was significantly different (Bray–Curtis PERMANOVA, *p* = 0.016). In addition, the composition of the microbiota in the samples of the LNK-/- group was more heterogeneous and significantly different from that of the WT group. The heat map showed that gut microbiota compositions between the LNK-/- and WT groups were markedly different (Figure 2B).

In the phylum-level taxonomy classification, the WT group was dominated by *Proteobacteria*, *Verrucomicrobia*, and *Bacteroidetes*; the LNK-/- group was dominated by *Bacteroidetes*, *Proteobacteria*, and *Firmicutes* (Figure 2C). Although bacteria are similar at the phylum level between the two groups, Figure 2C showed that their proportion was different. The WT group was dominated by *Proteobacteria* and had a relative abundance of *Verrucomicrobia*, with the significance compared with LNK-/- mice (*p* < 0.05) (Figure 2D), while the LNK-/- group has a relatively large proportion of *Firmicutes* (*p* < 0.05) and *Bacteroidetes* (Figure 2D).

According to the results of the linear discriminant analysis effect size (LEfSe) (LDA ≥ 2.0), the abundances of *Proteobacteria*, *Helicobacteraceae*, *Epsilonproteobacteria*, and *Campylobacterales* were significantly increased in WT mice, while the abundance of *Erysipelotrichales*, *Allobaculum,* and *Bacteroidales* was significantly increased in LNK-/- mice (Figure 2E).

To explore the gut microbial differences between LNK-/- and WT mice further, we used STAMP software to analyze the genera with significant differences (*p* < 0.05). We found that the abundances of some short-chain fatty acid (SCFA)-producing genera in the WT groups were significantly lower than in the LNK-/- groups, such as *Prevotella_9*, *Prevotellaceae_UCG-001*, *Clostridium_sensu_strict_1*, *Ruminococcaceae_UCG-010*, and *Stenotrophomonas* (Figure 2F).

## 4. Discussion

Our previous study showed that upon the consumption of HFD, LNK-/- mice had a loss of appetite, and WT mice accumulated more intrahepatic triglyceride, TG, and FFA compared with LNK-/- mice. LNK plays a pivotal role in adipose glucose transport by regulating insulin-mediated IRS1/PI3K/Akt/AS160 signaling. In this study, we found that the abundance of *Proteobacteria* was significantly increased in the WT mice group, which was one of the main LPS-producing bacteria. Some SCFA-producing genera in WT groups were significantly lower than in the LNK-/- groups.

LPS is also called endotoxin. The complex of LPS and its receptor CD14 can be recognized by Toll-like receptor 4 (TLR4) on the surface of immune cells to induce an inflammatory response. When the change in diet or the use of antibiotics affects the balance of gut microbiota, the number of harmful bacteria such as G- bacteria increases, and the decomposed product LPS passes into the blood circulation through the intestinal epithelium to cause endotoxemia, which triggers a systemic inflammatory response [18]. This study revealed that inflammation and LPS levels were elevated in patients with type 2 diabetes. Both animal and human experiments have demonstrated that the direct injection of LPS can increase fasting blood glucose and insulin levels, resulting in hyperinsulinemia and insulin resistance. When the number of G- bacteria decreased with the use of antibiotics, the amount of LPS entering the circulation decreased, which could relieve the systemic inflammation and increase insulin sensitivity. LPS receptor CD14 knockout mice fed a high-fat diet or injected with LPS had decreased inflammatory factors in adipose tissue, increased insulin sensitivity in liver and adipose tissue, and had a delayed development of insulin resistance, and their weight gain slowed down [19,20,21,22]. The results suggest that LPS plays an important role in the induction of the inflammatory response and insulin resistance.

The intestinal microbiota may affect the content of circulating LPS in the following two ways to induce insulin resistance. For one thing, the structure of intestinal microbiota is unbalanced, the number of G + bacteria is decreased, the proportion of G- bacteria is increased, and the production of LPS is increased. Studies have shown that the number of G + bacteria such as Clostridium decreased and the number of LPS-containing bacteria such as Bacteroides and Proteobacteria increased in diabetic patients. Adding Lactobacillus and Bifidobacterium to the diet of high-fat-induced obese mice could help restore a balance between probiotics and pernicious bacteria in the gut and increase insulin sensitivity. The addition of prebiotic oligosaccharides to a high-fat diet-induced diabetic mouse model also increased the number of bifidobacteria, decreased the level of LPS, and improved insulin secretion and inflammation, which was significantly associated with the number of bifidobacteria [23]. Additionally, intestinal microbiota alter intestinal permeability. Studies have shown that a high-fat diet may interact with intestinal microbiota, alter intestinal permeability, promote the rise of LPS levels, and cause an inflammatory state and insulin resistance [24,25]. The intestinal microbiota selectively regulates the expression of colonic Cannabinoid receptor 1, which affects intestinal permeability by altering the distribution of Claudin-1 [26]. In addition, obesity itself affects intestinal permeability. A study of normal-weight and overweight healthy women showed a positive correlation between gut permeability and waist circumference and visceral fat content [27]. Increased visceral adipose promotes the secretion of the pro-inflammatory factors TNF α, IL-1, and IL-6 by infiltrating macrophages in adipose tissue and reducing the production of the anti-inflammatory factor adiponectin. With the action of multiple pro-inflammatory factors, intestinal mucus production was decreased, and intestinal permeability was increased. TNF-α can also act on tight junction proteins, resulting in the increased permeability of the tight junction of intestinal cells [28,29,30]. These proinflammatory factors can also promote insulin resistance and lipid storage in adipocytes, thereby forming a vicious cycle.

Probiotics such as Bifidobacterium, Lactobacillus, and Prevotella_9 can promote the release of SCFAs from the undigested soluble dietary fiber in the colon via fermentation, at the same time reducing the intestinal pH, inhibiting the growth of harmful bacteria, to reduce the production of LPS in the intestinal lumen [31]. SCFAs can also promote the secretion of insulin by pancreatic β cells by regulating the secretion of gut-derived hormones, such as glucagon-like Peptide 1 (Glp-1), Glucagon Peptide 2 (Glp-2), Peptide YY (PYY), and glucose-dependent insulinotropic Peptide (GIP), etc., to increase insulin sensitivity and suppress appetite and food intake, thereby improving insulin resistance. After 8-week oral medication of VSL#3 probiotics containing 8 kinds of viable bacteria, the diet-induced obesity mice had increased GLP-1 production, decreased food intake, reduced body weight, and improved glucose tolerance. Their intestinal microbiota composition also changed the number of probiotics of Firmicutes such as lactobacillus, and Bifidobacterium increased, which was related to the increase in butyrate in SCFAs [32]. Butyrate can improve the function of the intestine, promote the activity of the intestine, and has a better therapeutic effect on patients with a loss of appetite, diarrhea, dyspepsia, and so on. In addition, butyrate can promote the reduction of dietary intake and digestion and is also beneficial to obese or fatty liver patients [32]. In addition, another study showed that healthy volunteers ate inulin-containing foods that promoted probiotic growth and a regular diet, respectively. Moreover, GLP-2 was found to be increased in fasting serum and decreased in intestinal permeability after eating inulin-containing foods [33]. The results demonstrated that probiotics could promote the production of SCFAs and the secretion of GLP-1 and Glp-2 by regulating the balance of intestinal microbiota, further improving intestinal permeability and alleviating IR.

Our research explored the changes in the gut microbiota in LNK-/- and ET mice, which provided new ideas for the mechanism and treatment of MetS and IR. Although previous studies had shown that the disorder of intestine microbiota was related to MetS, the underlying mechanism remains unclear. Therefore, this study was a supplement to this research field. Nevertheless, this study still had some shortcomings. Firstly, as is known to all, sex hormones strongly influence body fat distribution and adipocyte differentiation. Estrogen and testosterone differentially affect adipocyte physiology and estrogens play a leading role in the causes and consequences of female obesity. Therefore, in this study, to avoid the influence of estrogen on the occurrence of obesity, we did not put male and female mice together to compare, and only collected fecal samples based on previous obesity-induced IR male mouse models. The sample size was not large enough, and there may be bias in the results for female mice. The results of female mice and the potential effects of sex hormones on gut microbiota need further research. Secondly, in the study, we focused on the difference in gut microbiota between LNK-/- and WT mice. We will continue relevant studies, and the indexes such as LPS, butyrate, gut permeability, and mucosal structural changes will be measured or observed in our next study. The relationship between changes in gut microbiomes and IR needs to be confirmed by further experiments. Finally, the 16S rRNA gene sequencing had some limitations, such as a short reading length, sequencing errors, and difficulty in evaluating and operating taxa. It would be important to combine signaling pathways and metabolomics analysis in the next step.

## 5. Conclusions

Our research described that the structure and composition of the gut microbiota community between LNK-/- and WT mice were significantly different. The change in the gut microbial structure and composition of obese WT mice might aggravate glucolipid metabolic disorder and IR by increasing the production of LPS while reducing the production of SCFAs.

## Figures and Tables

**Figure 1 jcm-12-01767-f001:**
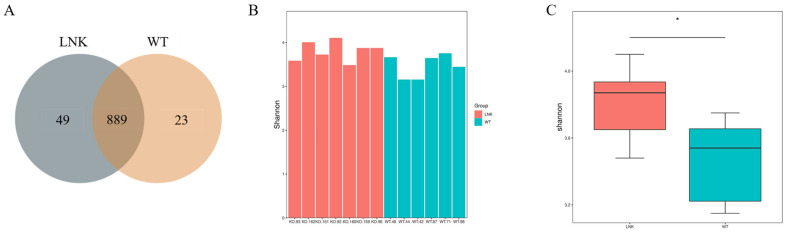
Diversity difference in the intestinal microbiota of LNK-/- and WT mice. (**A**) Venn diagram of common and specific intestinal microbe species of LNK-/- and WT group. (**B**,**C**) The α diversity of the gut microbiota between the LNK-/- and WT groups, calculated using the Shannon index (* *p* < 0.05).

**Figure 2 jcm-12-01767-f002:**
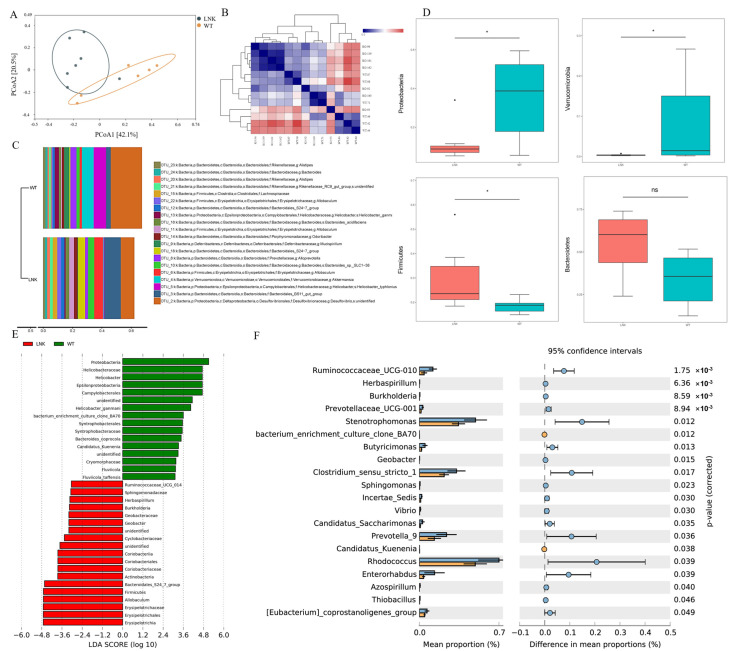
Composition and abundance difference in the intestinal microbiota between LNK-/- and WT mice. (**A**) Principal coordinate analysis (PCoA) of gut microbiota based on Bray–Curtis dissimilarity (Bray–Curtis PERMANOVA, *p* = 0.016). (**B**) The heat map of gut microbiota compositions between LNK-/- and WT group. (**C**) Taxonomic classification of the gut microbiota of LNK-/- and WT group at the level of phylum. (**D**) Boxplots of Proteobacteria, Verrucomicrobia, Firmicutes, and Bacteroidetes between LNK-/- and WT group (* *p* < 0.05). (**E**) Linear discriminant analysis of the differential abundance gut microbiota between LNK-/- and WT group. (**F**) Differences in bacterial genera between LNK-/- and WT group (*p* < 0.05).

## Data Availability

Not applicable.

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
