# Peer review of "Gut Microbiota Dysbiosis Ameliorates in LNK-Deficient Mouse Models with Obesity-Induced Insulin Resistance Improvement"

_jcm, 2023, doi:10.3390/jcm12051767_

Round 1
Reviewer 1 Report
The authors present data related to the, Gut microbiota dysbiosis ameliorates in LNK-deficient mouse model of obesity-induced insulin resistance improvement. This is a nice article. The few weaknesses are as follows and should be addressed by the author.
-
Did the authors measure the body weight of the animals?
-
If the authors did the same experiment on female animals, should those data be included in the paper? If not, please explain.
-
Did the authors measure LPS both in Wt and LNK-/- mice?
-
In LNK-/- mice, did the authors measure butyrate?
-
Have the authors collected colons and observed their architecture in both groups of animals? If the authors have data, please include them.
-
The results of Figure 2 need to be rewritten clearly
-
To understand LNK-/- mice's importance, IR and gut microbe signaling mechanisms must be included
-
LNK-/- and gut microbes need to be discussed in depth
-
In LNK-/- mice, have authors observed any changes in Insluin levels?
Reviewer 2 Report
Dear authors, below please find detailed comments that I hope will help improving the manuscript.
1. I discourage the use of the term gut flora or microflora, gut microbiota is preferable.
2. I understand that the data is part of a greater study regarding the metabolic role of LNK, however the results of previous works related to this topic are crucial to understand the current involvement of gut microbiota so I think that they need to be provided at least as supplementary data.
3. Overall, the scope of the research is interesting and results of microbiota analysis seems promising, but the mechanisms linking differences in microbiota composition and metabolic state of knockout and WT mice are not explored at all. LPS measurement in plasma, determination of SCFAs in fecal content, or expression of PRRs like TLR-4 or NLRP-3, are valuable data that could provide a better understanding of this relationship.
4. Presumably due to this lack of more analytical data, discussion is quite speculative and barely based in the current study results. Again, previous suggested determinations may help to improve discussion of the results.
5. Regarding diversity analysis, authors provide Simpson diversity index, as this index is inversely correlated with actual diversity is quite unintuitive for the reader so I strongly recommend calculate the inverse Simpson index or the Shannon diversity index which is considered to be more consistent in this type of data.
6. Lines 144-145, authors stated that LNK-/- mice showed less appetite that WT counterparts, was food intake measured across the experiment? Moreover, if there is indeed a different eating behavior between them, it may be interesting to determine levels of leptin.
7. Lines 159-160, a statistical test (PERMANOVA) results may be provided to sustain significant difference in microbiota composition between groups.
8. The role of the phylum Verrucomicrobia, which Akkermansia muciniphila belongs, has been extensively discussed in metabolic disease, as this taxa abundance is higher in WT mice, this question may be discussed.
9. Differences in microbiota composition at phylum level are one of the main findings of the study so I think that a box plot representation of relative abundance of these discriminating phyla should be provided in Figure 2 rather than the bar graph.
Round 2
Reviewer 2 Report
I want to thank the authors for their kind answers to my comments and for their effort to include some of my suggestions in the revised version of the manuscript.
Author Response
Thanks for the reviewer's comments a lot!